# Diagnostic Accuracy of Loop-Mediated Isothermal Amplification Assay for Group B Streptococcus Detection in Recto-Vaginal Swab: Comparison with Polymerase Chain Reaction Test and Conventional Culture

**DOI:** 10.3390/diagnostics12071569

**Published:** 2022-06-28

**Authors:** Ji-Hee Sung, Hyun-Hwa Cha, Nan-Young Lee, Won-Ki Lee, Yeseul Choi, Hyung-Soo Han, Yoo-Young Lee, Gun-Oh Chong, Won-Joon Seong

**Affiliations:** 1Department of Obstetrics and Gynecology, Samsung Medical Center, Sungkyunkwan University School of Medicine, Seoul 06351, Korea; jihee.sung@samsung.com (J.-H.S.); yooyoung.lee@samsung.com (Y.-Y.L.); 2Department of Obstetrics and Gynecology, Kyungpook National University Chilgok Hospital, Daegu 41404, Korea; chh9861@knu.ac.kr; 3Department of Obstetrics and Gynecology, School of Medicine, Kyungpook National University, Daegu 41404, Korea; 4Department of Laboratory Medicine, Kyungpook National University Chilgok Hospital, School of Medicine, Kyungpook National University, Daegu 41404, Korea; leenanyoung70@gmail.com; 5Department Medical Informatics, School of Medicine, Kyungpook National University, Daegu 41944, Korea; wonlee@knu.ac.kr; 6Clinical Omics Research Center, School of Medicine, Kyungpook National University, Daegu 41940, Korea; yeseul.choi830@gmail.com; 7Department of Physiology, School of Medicine, Kyungpook National University, Daegu 41405, Korea; hshan@knu.ac.kr

**Keywords:** group B streptococcus, diagnosis, loop-mediated isothermal amplification assay, polymerase chain reaction assay

## Abstract

A rapid method for obtaining group B streptococcus (GBS) screening results has been required in the obstetric field. We aimed to determine the diagnostic performance of the Loop-Mediated Isothermal Amplification (LAMP) assay is acceptable compared to the existing polymerase chain reaction (PCR) assay. The study involved 527 pregnant women aged 19 to 44 years. Rectovaginal swabs were collected between 35 and 37 weeks of gestation or prior to impending preterm births or term labor without GBS screening. We presented the diagnostic performance of the LAMP assay with a 95% confidence interval (CI) compared to the PCR and microbiological culture. In total, 115 (21.8%), 115 (21.8%) and 23 (4.4%) patients showed positive results using the LAMP, PCR assay and microbiological culture method, respectively. The LAMP assay showed 100% sensitivity (95% CI, 96.8–100.0), 100% specificity (95% CI, 99.1–100.0) and 100% diagnostic accuracy (95% CI, 99.3–100.0) with the reference being the PCR assay. Meanwhile, the LAMP assay showed 87.0% sensitivity (95% CI, 71.0–100.0), 81.2% specificity (95% CI, 77.6–84.7), and 81.4% diagnostic accuracy (95% CI, 78.0–84.8) with the microbiological culture as a reference. This study presented the LAMP assay as an acceptable method for GBS screening with a similar performance to the existing PCR method.

## 1. Introduction

*Streptococcus agalactiae*, also known as group B *streptococcus* (GBS), is a commensal Gram-positive bacterium that can transiently colonize the vagina, urethra, and lower gastrointestinal track [1]. While GBS is an asymptomatic colonizer of healthy adults, it can cause severe infection in neonates [2,3]. Because GBS is the leading cause of newborn infection, the Centers for Disease Control (CDC) implemented a universal guideline that recommends screening of all pregnant women at 35–37 weeks of gestation for GBS and administering intrapartum antibiotic prophylaxis (IAP) in pregnant women who test positive [2,3]. Until recently, the standard method for maternal GBS carriage detection was microbiological culture. However, microbiological culture has a long turnaround time and its sensitivity is only 54–87% [4]. Moreover, vaginal GBS colonization can be intermittent during pregnancy [5]; at least 10% of women who tested negative using late antenatal culture screening were found to be positive at the time of delivery [6]. Therefore, it is recognized that a rapid, sensitive, and specific GBS test may have benefit during the intrapartum period or following the rupture of membranes.

The loop-mediated isothermal amplification (LAMP) method can amplify target nucleotide sequences at isothermal conditions (usually 60–65 °C) within 90 min using four or six primers [7,8]. Amplification can be detected by macroscopic observation of turbidity or color change of a fluorescent dye [8]. There are several commercial GBS molecular tests using polymerase chain reaction (PCR) for intrapartum screening. However, PCR tests require equipment and reagents that may not be available in all microbiology laboratories and are more expensive than those needed for cultures. Because the LAMP assay is a rapid, practical, and relatively straightforward method, it may be useful for rapid testing, i.e., point-of-care intrapartum GBS screening. On these backgrounds, the objective of this study was to evaluate the diagnostic performance of LAMP assay (Isopollp^®^ easy GBS Detection Kit) for detection of GBS in maternal recto-vaginal swabs and to compare it with PCR testing (BD MAX™ System) and microbiological culture.

## 2. Materials and Methods

### 2.1. Study Design and Participants

This study was performed between June 2018 and November 2021. Pregnant women between 35 and 37 weeks of gestation or who were expecting impending preterm delivery or term labor without GBS screening in two tertiary academic hospitals were included in this study. Each patient was swabbed with lower one-third vaginal, perineal, and rectal specimens collected on three sterile cotton swabs simultaneously by obstetric experts. One swab was used for microbiological culture and the remaining two swabs were preserved at −70 °C. The stored specimens were used for the BD MAX™ System and LAMP assay later. We regarded the BD MAX™ System results as reference results, and the laboratory medicine expert performed the LAMP assay with blinded reference results.

### 2.2. Culture of GBS

The vaginal/rectal swab specimens were incubated in BBL™ Lim broth enrichment media (Cat. No. 296266, Becton Dickinson, Franklin Lakes, NJ, USA). The broth tubes were incubated at 35 to 37 °C in ambient air or 5% CO_2_ for 18–24 h, and then the broth was sub-cultured onto 5% sheep blood agar plate (BAP, ASAN PHARM. CO., LTD., Seoul, South Korea) according to the established CDC guidelines [3]. The β-hemolytic or non-hemolytic colonies were suspected as GBS based on Gram stain and catalase test. The identification of a colony was confirmed using the Vitek2 Gram-positive identification system (bioMérieux, Marcy l’Etoile, France) and matrix-assisted laser desorption ionization–time of flight mass spectrometry (MALDI-TOF MS, Bruker Daltonics, Billerica, MA, USA). 16S rRNA sequencing was also performed via the ABI 3730xl DNA analyzer (Applied Biosystems, Foster City, CA, USA) to confirm the exact identification of bacteria.

### 2.3. BD MAX™ GBS Assay (Cat. No. 441772, Becton Dickinson)

The vaginal/rectal swab specimen was used for the Lim broth culture experiments, which was inoculated into a BBL™ Lim broth (BD Diagnostic Systems, Sparks, MD, USA) and incubated at 35 to 37 °C in ambient air for 18–24 h. The enrichment broth was then used for BD MAX or LAMP assay. The 24 cultured broths (15 μL) were mixed with BD MAX GBS sample preparation reagent and processed on the BD MAX™ system (a real-time PCR) using the BD MAX GBS assay according to the manufacturer’s instructions. The amplified targets were detected in real time using Scorpions^®^ chemistry-based fluorogenic oligonucleotide probe molecules specific to the amplicons for the respective targets. Briefly, the BD MAX™ system automatically extracts the nucleic acid using a combination of heat, lytic enzymes, and magnetic capture beads. The BD MAX GBS PCR amplifies a section of the *cfb* gene target sequence of the GBS chromosome.

### 2.4. LAMP Assay Using MmaxSureTM EZ GBS Detection Kit (Cat. No. 52313, Mmonitor, Daegu, South Korea)

For GBS DNA extraction, the cultured broth samples (20 μL) were mixed with Mmaxpress^®^ GL01 Prep Kit (Cat. No. 92205, Mmonitor) and incubated for 5 min at 63 °C. The GBS DNA extracts (5 μL) were added into MmaxSure™ EZ GBS Detection Kit (Mmonitor), and then each sample was amplified for 40 min at 63 °C and inactivated for 2 min at 80 °C to amplify the *sip* gene encoding the Sip surface immunogenic protein that is produced by all GBS isolates. The result was identified by color change. As a result of the reaction, if the color is blue, the result is considered positive. If the color changes to purple, the test is negative (Figure 1).

### 2.5. Statistical Analysis

For the clinical validation, we defined the BD MAX™ GBS assay, which was approved for use in antenatal GBS screening, as the reference result. Sensitivity, specificity, and diagnostic accuracy were presented with a 95% confidence interval. Furthermore, we presented the diagnostic performance of the LAMP assay compared to microbiological culture. Statistical analysis was performed using SAS version 9.4 (SAS Inc., Carry, NC, USA).

### 2.6. Ethical Consideration

This study was approved by the Institutional Review Board of Kyungpook National University Chilgok Hospital and Samsung Medical Center (IRB file No.: KNUCH-2020-09-001-002, 2020-09-109-016), and written informed consent was obtained from participants.

## 3. Results

### 3.1. Basal Characteristics

The mean maternal age was 32.4 ± 4.2 years and the median gestational age at sampling and gestational age at delivery were 36.5 (22.6–39.4) and 38.6 (24.4–41.1) weeks of gestation, respectively. Out of a total of 527 patients, 115 (21.8%), 115 (21.8%) and 23 (4.4%) patients showed positive results by BD MAX™ GBS Assay, LAMP assay and microbiological culture method, respectively.

### 3.2. LAMP Assay vs. BD MAX™ GBS Assay

LAMP assay results revealed 100% consistency with the BD MAX™ GBS Assay as shown in Table 1. When we defined the BD MAX™ GBS Assay as the reference result, the LAMP assay showed a sensitivity of 100% (95% CI, 96.8–100.0), specificity of 100.0% (95% CI, 99.1–100.0), and diagnostic accuracy of 100.0% (95% CI, 99.3–100.0).

### 3.3. LAMP Assay vs. Microbiological Culture

LAMP assay results revealed 81.4% consistency with the microbiological culture (Table 2). When we defined the microbiological culture as the reference result, the LAMP assay showed a sensitivity of 87.0% (95% CI, 71.0–100), specificity of 81.2% (95% CI, 77.6–84.7), and diagnostic accuracy of 81.4% (95% CI, 78.0–84.8).

In addition, we found that the PCR method reported an additional 95 positive cases, which showed negative conventional culture results. Interestingly, there were three cases with positive biological culture results that were not detected by the PCR method.

## 4. Discussion

In this study, we demonstrated that the LAMP assay showed 100% diagnostic accuracy compared to the BD MAX™ System, which is an approved real-time PCR method for GBS screening in pregnant women. In other words, the LAMP method is not inferior to that of an approved PCR test. In addition, we found that the LAMP assay showed acceptable sensitivity and specificity considering microbiological culture as the reference. Most cases (95/98) in which PCR and microbiological culture results were inconsistent were culture-negative and PCR-positive results. Interestingly, three cases with culture-positive and PCR-negative results exhibited heavy colonization on culture. Because there was no evidence of congenital infection in neonates from these cases, we carefully presumed that these cases might have been contaminated. However, there is still possibility of detection limit of PCR method. Contrarily, there were 95 cases with inconsistent results, which were culture-negative and PCR-positive results, in our study group. We propose to pay attention to cases with inconsistent results, as a report has shown that 60% of GBS early-onset disease (EOD) in neonates had a negative rectovaginal GBS screening results between 35 to 37 weeks of gestation [2]. There is a possibility that only samples with abundance of GBS can be detected by microbiological method. The studies regarding the neonatal outcomes in cases with inconsistent results are warranted.

Generally, GBS EOD has been regarded as a Western countries’ disease. Therefore, Korean obstetricians had little interest in GBS screening and performed IAP based on the presence of risk factors such as preterm birth before 37 weeks of gestation, amniotic membrane rupture for 18 h or more, or intrapartum fever (38.0 °C or higher). However, several studies about the prevalence of GBS colonization in Korean pregnant women reported a colonization rate ranging between 1.96% and 11.5% that has been increasing during the last three decades [9,10,11,12,13]. These reports have shown different colonization rates depending on gestational age at screening, culture media, and study period. The GBS colonization rate has been reported to increase with advanced gestational age at screening [14]. Most studies regarding GBS colonization rate included pregnant women beyond 35 weeks of gestation unlike our study that included preterm labor before 35 weeks of gestion. We thought that is the reason our study showed a 4.4% colonization rate, which is relatively lower than that in previous reports. However, the latest report regarding the prevalence of GBS colonization from a single tertiary care included pregnant women beyond 20 weeks of gestation showed an 11.6% colonization rate [9]. A standard culture method using Lim broth enrichment with subsequent culture on 5% blood agar media showed higher GBS colonization prevalence rate compared to direct culture [15]. The colonization rate based on PCR methods in our study group was similar with previous studies; however, it was lower by microbiological culture even though we used a standard culture method. These meant GBS colonization rate by microbiological culture method would vary depending on each institution even in the same country.

Meanwhile, nucleic acid amplification testing (NAAT) for GBS detection is a rapid test performed at the time of presentation in labor or for women at term who have an unknown or unavailable antepartum GBS screening test that can provide rapid results. However, a 1–2 h turnaround time of NAAT does not permit the full enrichment broth incubation step that is required to maximize positive results [16]. Therefore, CDC guidelines recommend IAP in intrapartum women with risk factors for GBS EOD even if they return negative intrapartum NAAT results [3]. Recently, several FDA-cleared NAATs for GBS detection were performed on 18–24 enrichment broth culture for increasing the sensitivity and specificity [16]. The GBS BD MAX™ System is a PCR test intended for use with enriched Lim broth culture after incubation of swab samples for more than 18 h. This is similar to several FDA-cleared NAATs performed on 18–24 h enrichment broth cultures aimed at increasing sensitivity and specificity. These can provide results for 24 specimens in approximately 2.5 h while culture-based methods can take from 48 to 72 h for a final result after enrichment. Results can be maximized by performing incubation on Lim broth; however, this method requires a real-time PCR machine and nucleic acid purification system. The LAMP assay using MmaxSureTM EZ GBS Detection Kit, which was used in this study, had good sensitivity and specificity for detection of GBS and was reliable compared to the existing PCR method. Moreover, it required shorter turnaround time (60–80 min after 18 h incubation) and simple equipment (heat block for 63 °C and simple nucleic purification solutions). Previous reports have shown other benefits of LAMP assay [17,18,19]. Gonzalez et al. demonstrated that colorimetric LAMP assay was a quantitative method comparable in RT-qPCR that can detect SARS-CoV-2 by color changes correlating with viral copy number even at low viral copy numbers [17]. Chung et al. also developed a workflow of single-cell reverse transcription LAMP (scRT-LAMP) to quantify mRNA expression levels in different cell types [18]. In addition, the LAMP method can provide rapid diagnosis of bacterial infection during management of intensive care unit patients who were required to use precise antibiotics [19].

Until now, the microbiological method remains the gold standard screening method for GBS colonization to reduce EOD in neonates. However, the majority of EOD by GBS occurred in infants whose mothers tested negative for GBS colonization since the colonization status for GBS during pregnancy can change [20,21]. In addition, there was a considerable number of pregnant women who delivered their babies without antepartum GBS screening considering the prevalence of preterm birth. Therefore, development and implementation of rapid and accurate methods for GBS screening is required. We think our approach may help in solving this problem.

## 5. Conclusions

The LAMP assay could be used in the identification of intrapartum GBS prophylaxis candidates who presented in labor without antepartum GBS result.

## Figures and Tables

**Figure 1 diagnostics-12-01569-f001:**
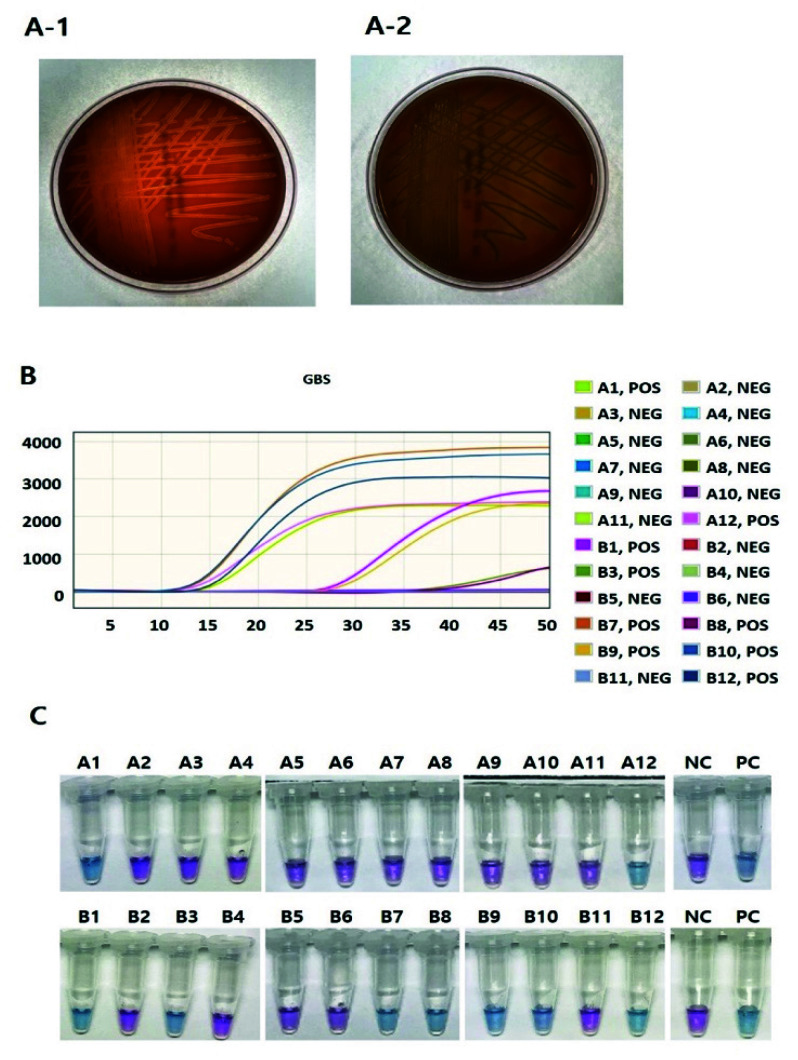
Representative results of the Culture, BD MAX™ and LAMP assay for the presence of GBS in the cultured broth samples. (**A-1**) Sub-cultured colonies on blood agar plate after Lim Broth enrichment for Group B *Streptococci.* (**A-2**) Sub-cultured colonies that are not suspected as Group B *Streptococci.* (**B**) BD MAX™ GBS Assay. The 9 positive samples are shown in the graph, the 15 negative samples are not shown in graph. POS: positive; NEG: negative (**C**) LAMP Assay. The 9 positive samples are shown as blue color, the 15 negative samples are shown as purple color. PC: positive control; *NC*: negative control; Purple: negative; Blue: positive.

**Table 1 diagnostics-12-01569-t001:** Diagnostic performance of Loop-Mediated Isothermal Amplification (LAMP) assays with reference to BD MAX™.

Group B Streptococcus (GBS) Diagnosis		BD MAX™ GBS Assay		Total
		Positive	Negative	
**LAMP assay**	Positive	115	0	115
	Negative	0	412	412
**Total**		115	412	527
**Sensitivity (%, 95% CI) ***		100 (96.8–100.0)		
**Specificity (%, 95% CI) ***			100 (99.1–100.0)	
**Diagnostic accuracy (%, 95% CI) ***				100 (99.3–100.0)

* Wald’s with continuity correction.

**Table 2 diagnostics-12-01569-t002:** Diagnostic performance of LAMP assay with reference to microbiological culture.

GBS Diagnosis		Microbiological Culture		Total
		Positive	Negative	
**LAMP assay**	Positive	20	95	115
	Negative	3	409	412
**Total**		23	504	527
**Sensitivity (%, 95% CI) ***		87.0 (71.0–100.0)		
**Specificity (%, 95% CI) ***			81.2 (77.6–84.7)	
**Diagnostic accuracy (%, 95% CI) ***				81.4 (78.0–84.8)

* Wald’s with continuity correction.

## Data Availability

The data presented in this study are available on request from the corresponding author.

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
