# Peer review of "Diagnostic Accuracy of Loop-Mediated Isothermal Amplification Assay for Group B Streptococcus Detection in Recto-Vaginal Swab: Comparison with Polymerase Chain Reaction Test and Conventional Culture"

_diagnostics, 2022, doi:10.3390/diagnostics12071569_

Round 1

Reviewer 1 Report

In this manuscript, the authors evaluate the diagnostic accuracy of Group B Streptococcus Detection in Recto-Vaginal Swab of 527 human samples with 1. Loop-Mediated Isothermal Amplification (LAMP) Assay, 2. Polymerase Chain Reaction Test, and 3. Conventional Culture Method. The authors have addressed the diagnostic performance with sensitivity and specificity, accuracy, together with analysis with confidence interval (CI)). In general, it is one of the study highlighting the advantages of LAMP that provide similar performance to exisiting PCR method. There is some unclear information in this paper, that the authors should address prior to acceptance into publication.

1. In Figure 1A, the authors should put the negative result of sub-cultured colony and indicate how one determines POS/NEG based on the result. In Figure 1B, the label of A1-B12 with the color-coding in the real-time curve is unclear. There was no indication of the sample corresponding to the A1-B12 in both Figures 1B and 1C. Why is there a separation between A1-A4, A5-A8, and A9-A12, etc

2. It is confusing why the authors put LAMP assay as different position in comparison, i.e. why LAMP assay is put after vs. in comparison to microbiological culture. Since the evaluation of the diagnostic performance of the LAMP assay is the objective of this paper, it will be better to keep the LAMP assay in front in comparing other standard assays. In this case, we advise the authors to revise the table to the correct order.

3. The authors have discussed the advantages of LAMP, such as a shorter turnaround time, apart from good sensitivity and specificity. The authors should also discuss other benefits, such as the ability for low copy number (e.g. virus) quantification in coloimetric LAMP (Anal. Methods, 2021,13, 169-178; DOI https://doi.org/10.1039/D0AY01658F), detection in single cell (Lab Chip, 2019,19, 2425-2434; https://doi.org/10.1039/C9LC00161A), integration into microfluidics (Biosensors & Bioelectronics, 2017, 93:212-219; https://doi.org/10.1016/j.bios.2016.09.001), and digital-based quantification (Sci Rep, 2017 7, 14586; https://doi.org/10.1038/s41598-017-14698-x) in order to convince the readers of the potential of LAMP in place of the current standard assays used extensively over the world.

Reviewer 2 Report

In their manuscript entitled "Diagnostic Accuracy of Loop-Mediated Isothermal Amplification Assay for Group B Streptococcus Detection in Recto-Vaginal Swab: Comparison with Polymerase Chain  Reaction Test and Conventional Culture", Sung and colleagues describe a Loop-Mediated Isothermal Amplification (LAMP) assay as an alternative for already existing commercial PCR methodologies for the rapid screening of group B streptococcus (GBS) in pregnant women. The authors performed a total of 527 samples and analysed them by  the traditional culture methodologies, using the commercial PCR-based  BD MAXTMAssay, and the here proposed  LAMP assay. The authors conclude that  the LAMP assay has a sensitivity , specificity and diagnostic accuracy similar to that of the reference PCR BD MAXTM assay.

The manuscript in general well organized and written in a comprehensive way. There are however some issues that the authors should address:

1) in the abstract, the sentence "The LAMP assay showed 100% sensitivity 33 [95% CI; 96.8-100.0], specificity [95% CI; 99.1-100.0] and diagnostic accuracy [95% CI; 99.3-100.0] 34 with the reference being the PCR assay." should be more precise, clearly indicating that the performance of the LAMP method is similar to the BD MAXTM assay.

2) Although in line 112 it is described that the LAMP procedure was designed to amplify the the sip gene, no information on the primers used is provided. This information should be included in the corresponding part of the Materials and Methods section.

3) Figure 1: The legend should appear below the image and not before.

4) line 103: " a section of the cfb gene". The "cbf" should be in italics.

5) line 117: correct "Sub-culutred colony", as "culutred" should be cultured.

6) The sentence in lines 185-186 " Later gestational age at delivery, and Lim broth, blood agar media seemed to be associated with higher prevalence of GBS" seems strange. How do Lim broth and blood agar media contibute to a higher prevalence of GBS? Better diagnosis? Rephrase.

7) line 213: "...and simple nucleic purification solutions". Nucleic should be nucelic acids.

8) in the Discussion section, the authors should better address the significance of PCR-positive results and culture-negative results. Most probably this issue has to to with the relative abundance of GBS in the sample. Positive PCR and negative culture results turn into cases of disease?

Reviewer 3 Report

This manuscript compares the diagnosis of group B streptococcus using culture and real-time PCR against the Loop-Mediated Isothermal Amplification (LAMP) assay.

The following comments are made:

1. Line 32. What was the percentage of positives with LAMP?

2.Lines 89-93. Did you perform all three tests for each strain? Is one confirmation enough? Why did you do all three?

3. Line 101. Mention that it is a real-time PCR and make a better description of the method to be able to compare its process with the other method and see the difficulties or facilities.

4. Lines 106-115. How did you do the test? You do not clearly mention how it is done, since it is the important part of their work and it is necessary to know the process in detail to see the ease of the method.

5. Lines 116-126. The foot of Figure 1, put it under the figure.

6. Lines 120-121. They must go in material and methods.

7.Figure 1B. It is difficult to see the colors, put a table. Or are you just trying to show what the essay looks like? If so, indicate it in the figure caption.

8. Lines 134-137. They are ethical considerations, not statistics. Put in another section.

9. Lines 143-145. “115 (21.8%) and 23 (4.4%) patients showed positive results by PCR methods (BD MAXTM GBS Assay, LAMP assay) and microbiological culture method, respectively”. You have two results for three methods, could you explain it? You should mention that you obtained the same PCR and LAMP value, so as not to confuse the reader.

10. Table 2. Correct the “Total” values ​​of the horizontal line.

11. Put what the * in Table 2 means.

12. Line 168. Indicate that it is real-time PCR.

13.Line 174-176. They were also negative for LAMP?. In the culture, as there is enrichment, it can be positive. Therefore, you should also review the detection limit of PCR and LAMP. Another option, did you confirm the culture was GBS?

14. Line 213. Compare the time of the three trials.

15. Line 223. What test?

Round 2

Reviewer 2 Report

The manuscript was revised taking into consideration the criticisms raised and therefore this reviewer has no further criticisms.

Author Response

Thank you for your kind comments 

Reviewer 3 Report

Lines 122-130 refer to how the tests were carried out, so you should be placed in the Material and Methods section. It's an important part of your job. A shorter description can be found in the figure caption.

Author Response

Thank you for your comment. As your point, we moved lines 122-130 to Materials section. Also we modified this part accordingly and simplified the figure legend. We pointed the revised part as green color. 
